# Flagellin-Fused Protein Targeting M2e and HA2 Induces Innate and T-Cell Responses in Mice of Different Genetic Lines

**DOI:** 10.3390/vaccines10122098

**Published:** 2022-12-08

**Authors:** Liudmila A. Stepanova, Marina A. Shuklina, Kirill A. Vasiliev, Anna A. Kovaleva, Inna G. Vidyaeva, Yana A. Zabrodskaya, Alexandr V. Korotkov, Liudmila M. Tsybalova

**Affiliations:** 1Smorodintsev Research Institute of Influenza, Russian Ministry of Health, 15/17 Prof. Popova Str., St. Petersburg 197376, Russia; 2Institute of Biomedical Systems and Biotechnology, Peter the Great St. Petersburg Polytechnic University, 29 Ulitsa Polytechnicheskaya, St. Petersburg 195251, Russia

**Keywords:** influenza, universal vaccine, M2e ectodomain, HA2, flagellin, recombinant protein, T-cell response

## Abstract

Efficient control of influenza A infection can potentially be achieved through the development of broad-spectrum vaccines. Recombinant proteins incorporating conserved influenza A virus peptides are one of the platforms for the development of cross-protective influenza vaccines. We constructed a recombinant protein Flg-HA2-2-4M2ehs, in which the extracellular domain of the M2 protein (M2e) and the sequence (aa76-130) of the second subunit of HA (HA2) were used as target antigens. In this study, we investigated the ability of the Flg-HA2-2-4M2ehs protein to activate innate immunity and stimulate the formation of T-cell response in mice of different genetic lines after intranasal immunization. Our studies showed that the Flg-HA2-2-4M2ehs protein was manifested in an increase in the relative content of neutrophils, monocytes, and interstitial macrophages, against the backdrop of a decrease in the level of dendritic cells and increased expression in the CD86 marker. In the lungs of BALB/c mice, immunization with the Flg-HA2-2-4M2ehs protein induced the formation of antigen-specific CD4+ and CD8+ effector memory T cells, producing TNF-α. In mice C57Bl/6, the formation of antigen-specific effector CD8+ T cells, predominantly producing IFN-γ+, was demonstrated. The data obtained showed the formation of CD8+ and CD4+ effector memory T cells expressing the CD107a.

## 1. Introduction

The limitations of seasonal vaccines and the threat of the emergence of pandemic influenza A viruses indicate that the development of universal influenza vaccination strategies remains a priority. Universal (cross-protective) vaccines should provide protection against different subtypes of influenza A viruses, including any future strains arising from antigenic shift or drift. Ideally, such a vaccine should provide 75% protection against influenza A viruses of phylogenetic groups one and two with a duration of protection of at least 1 year [1]. The most effective strategy for developing vaccines with a wide cross-protective potential is the formation of both humoral and T-cell responses [2].

Recombinant proteins incorporating conserved influenza A virus peptides are one of the platforms for the development of cross-protective influenza vaccines. The design and research of such vaccines (including clinical trials) is conducted by many scientific centers and companies that produce immunobiological preparations [3,4,5,6,7,8,9]. Previously, we constructed a recombinant vaccine protein Flg-HA2-2-4M2ehs, in which the extracellular domain of the M2 protein (M2e) and the sequence (aa76-130) of the second subunit of HA (HA2) were used as target antigens. Its immunogenicity (in terms of the production of specific local and systemic antibodies) and the breadth of its protection when administered intranasally and parenterally, were shown [10,11,12].

The extracellular peptide of M2 protein (24aa) is highly conserved. The interaction of the M2e peptide with the effectors of the immune system occurs on the surface of infected cells, on which it is presented in large quantities, while on the viral particle M2e it is represented by two or three units and is shielded by large surface proteins HA and NA, due to which it is inaccessible to immune effectors. Anti-M2e antibody eliminate influenza virus-infected cells due to the complement-dependent cytolysis, antibody-dependent cellular cytotoxicity, and/or antibody-dependent cellular phagocytosis. [13,14,15,16]. M2e-specific T cells can also mediate protection against influenza infection [15].

The second subunit of HA (HA2), which is responsible for the fusion of the viral and cell membranes in endosomes, thus ensuring the entry of the ribonucleic complex into the cytoplasm, is quite conserved within each phylogenetic group and has the potential for heterosubtypic protection [17,18]. It is also known that vaccines with HA2 of phylogenetic group two influenza A viruses can induce a stronger immune response than HA2 of the phylogenetic group one viruses [19]. The study of conserved HA2 epitopes of influenza A viruses belonging to the first and/or second phylogenetic group (aa 38-59, 23-185, 1-172, 76-103, 35-107) and the construction of recombinant proteins based on them showed that such proteins form both humoral and cytotoxic T-cell response and are effective in protecting against infection with lethal doses of homologous and heterologous viruses [20,21,22,23,24,25].

In this work, we investigated the ability of the recombinant protein Flg-HA2-2-4M2ehs to activate innate immunity and stimulate the formation of a T-cell response in mice of different genetic lines after intranasal immunization.

## 2. Materials and Methods

### 2.1. Recombinant Protein

The recombinant protein was designed and obtained at the Smorodintsev Research Institute of Influenza in collaboration with the Federal Research Center “Fundamentals of Biotechnology” Russian Academy of Science [10,11]. The recombinant protein (Flg-HA2–2-4M2ehs) was previously described by us [10,11], and included four copies of the M2e peptide and fragment (HA2-2, aa76-130) of the second subunit of HA influenza viruses of the second phylogenetic group, linked to the C-terminus of flagellin. The E. coli producing strain was obtained by transforming E. coli cells of the DLT1270 strain with the recombinant plasmid pQE30 Flg_H2_4Mehs. The recombinant protein was purified by metal-affinity chromatography.

### 2.2. Mice

The mice (female BALB/c (H-2d) and C57Bl/6 (H-2b), 16–18 g) were obtained from a certified laboratory animal nursery, the Stolbovaya mouse farm at the State Scientific Center of Biomedical Technologies, Russian Academy of Medical Sciences. The mice were kept in the vivarium at the Smorodintsev Research Institute of Influenza (RII) under existing regulations. The experiments were conducted in accordance with the Bioethics Committee of RII (protocol numbers: 07/01/19 and 19/01/20).

### 2.3. Immunization

For the phenotyping of innate immunity cells, C57Bl/6 mice (*n* = 15) were intranasally immunized once with the recombinant protein Flg-HA2-2-4M2ehs at a dose of 10 μg (20 μL)/mouse. The control group of mice (*n* = 20) was administered with 20 μL of PBS once.

To study the T-cell response, mice of two genetic lines (C57Bl/6 and BALB/c) were immunized with the recombinant protein Flg-HA2-2-4M2ehs intranasally (i/n) three times with an interval of 2 weeks at a dose of 10 μg/20 μL. The control group of mice was administrated i/n with 20 μL of PBS.

### 2.4. Collection of Mouse Lungs

To assess the dynamics of innate immunity cells, the mice were euthanized (cervical dislocation) and lungs were harvested from five mice before administration (control group of mice only) and 12, 24, and 48 h after immunization (from five mice of each group). To study the adaptive T-cell response, the lungs from the control and immunized mice were harvested 2 weeks after the third immunization following euthanasia (cervical dislocation).

A suspension of mouse lung cells was obtained as previously described [10]. In brief, the lungs in RPMI-1640 with 0.5 mg/mL collagenase (Sigma, C2674) and 25 µ/mL DNase (Sigma, D4263) were homogenized and then placed in a shaker for 45 min at 37 °C. The cell debris was removed by filtration (a pore diameter of 70 μm). ACK buffer (0.15 M NH4Cl, 1.0 M KHCO3, 0.1 mM Na2EDTA, pH 7.2–7.4) was used for red blood cell lysis. Then, the lung cells were washed, and the concentration of cells was adjusted to 5 × 10^6^ cells/mL.

### 2.5. Phenotyping of Innate Immunity Cells Using Multiparameter Flow Cytometry

The phenotyping of innate immunity cells (2 × 10^6^ cells/100 μL) was performed using a panel of fluorochrome-conjugated antibodies, including CD11b-PE/Cy7, CD11c-PE, MHCII-Alexa488, Ly-6G-PerCP-Cy5.5, CD45-APC/Cy7, CD64-BV421, and CD24-BV510 (Biolegend, San Diego, CA, USA). This panel enabled the alveolar macrophages (MHCII+CD64+CD11c+CD11b-), interstitial macrophages (MHCII+CD64+CD11b+CD11c+/−), monocytes (MHCII-CD64+CD24+), neutrophils (SSChiCD45+Ly6G+), and two dendritic cell populations: DC1 CD11b- (CD45+CD11c+CD11b-MHCII+CD64-CD24+) and DC2 CD11b+ (CD45+CD11c+CD11b+MHCII+CD64-CD24+) to be identified. To reduce the likelihood of nonspecific staining, a TrueStain reagent (Biolegend, San Diego, CA, USA) was added. The activation of innate immune cells was assessed based on the median fluorescence intensity (MFI) of the CD86-BV421 marker (Biolegend, San Diego, CA, USA). The results were recorded using a Cytoflex flow cytometer (Beckman Coulter, Brea, CA, USA). An analysis of flow cytometry data was performed using Kaluza 2.2 software (Beckman Coulter).

To assess the relative content of the cells of the studied populations in the lungs, the gating method proposed by Yu et al., 2016 [26], was used. This approach was originally developed to identify the main populations of innate immunity cells in the lungs and other organs of mice (small intestine, heart, kidneys, liver, peripheral organs of the immune system, etc.). A description of the gating tactics with an indication of the phenotype of the studied populations is shown in Appendix A.

### 2.6. Intracellular Cytokine Staining (ICS) Assay

The formation of an antigen-specific T-cell response in the lungs of mice was assessed by the number of effector memory CD4+ and CD8+ T cells (CD44+/CD62L-) and their functional characteristics by the expression of a degranulation marker (CD107a+) and the production of intracellular cytokines (TNFα and IFN-γ). ICS assay was performed as described earlier [10]. The cells (1 × 10^6^) were stimulated for 6 h at 370 °C with 10 μg of M2eh peptide and for 24 h with 1 μg of influenza virus in the presence of 1 μg/mL of Brefeldin A (BD, Franklin Lakes, NJ, USA) and purified hamster anti-mouse CD28. Furthermore, simultaneously with the activation, staining with the surface marker CD107a+ was carried out. After activation, the cells were transferred to conical-bottom plates. Then, the cells were washed, and Fc receptors were blocked with CD16/CD32 antibodies (Mouse BD Fc Block, BD, Franklin Lakes, NJ, USA). To identify living cells, we used Zombie Aqua (Zombie Aqua™ Fixable Viability Kit, Biolegend, San Diego, CA, USA). For the phenotyping of T-cell populations, the cells were stained with antibodies (CD3e-FITC, CD8a-APC-Cy7, CD4-PerCP, CD62L-PE-Cy7, and CD44-APC) and were permeabilized with the BD Cytofix/Cytoperm Plus (BD Bioscience, Franklin Lakes, NJ, USA) protocol. To assess the functional activity of the stimulated cells, staining was performed with anti-TNF-α-BV421, anti-IFN-γ-PE, and IL-2-BV711 (BD, Franklin Lakes, NJ, USA). A sample acquisition (150,000 live CD3+ were collected) was performed with a Cytoflex flow cytometer (Beckman Coulter) and analyzed using Kaluza, version 2.2 (Beckman Coulter, Brea, CA, USA). An initial gating of the studied cell population was performed in the FSC-A (abscissa) and FSC-H (*y*-axis) coordinates (gate S) in order to exclude cell aggregates from the analysis (Appendix A).

### 2.7. Statistical Analysis

In order to obtain quantitative data, the error of the mean and median (populations of T cells and innate immunity cells) was calculated. Statistical processing of innate immunity data was performed using the RStudio program. To compare the groups, the Student’s test (*t*-test) was used. An analysis of flow cytometry data was performed using Kaluza 2.2 software (Beckman Coulter). Differences in antigen-specific cytokine-producing T cells were evaluated by the Student’s test (*t*-test) and Holm–Sidak method for multiple comparisons. All data were checked for normal distribution by the Shapiro–Wilk test. If the *p* value was less than 0.05, the difference was considered to be significant. The analysis was carried out by using GraphPad Prism, version 6.0. A graphical presentation of the innate immunity and specific T-cell response data was produced in the form of Tukey plots, which display the median, lower, and upper quartiles as well as the minimum and maximum values of the sample and outliers.

## 3. Results

### 3.1. Dynamics of the Main Populations of Innate Immunity Cells in the Lungs

In order to study the dynamics of the population composition of innate immunity cells, C57Bl/6 laboratory mice were immunized intranasally with the Flg-HA2-2-4M2ehs protein at a dose of 10 µg/mouse. Then, mice were euthanized 12, 24, and 48 h after immunization, and their lungs were isolated. The control group of mice was administered intranasally with 20 μL of PBS and was also euthanized 0, 12, 24, and 48 h after administration in order to obtain the lungs. Populations of neutrophils, monocytes, alveolar macrophages, interstitial macrophages, and dendritic cells of two types (DC1CD11b- and DC2CD11b+) were phenotyped in a suspension of lung cells.

The results of the assessment of the relative content of the main populations of innate immunity cells at different times after immunization are presented in Figure 1A. Compared with the control group, mice immunized with Flg-HA2-4M2ehs showed a significant increase in the content of neutrophils at all studied points after drug administration. The most pronounced differences were noted 12 h after immunization (*p* = 0.00004), where the relative content of neutrophils in the lungs gradually decreased. The level of monocytes, on the contrary, was highest in the group of immunized animals after 48 h (*p* = 0.01). Additionally, at this time, a significant increase in the relative number of interstitial macrophages (*p* = 0.02) and a decrease in the level of alveolar macrophages were found. A decrease in the proportion of dendritic cells (CD11b- and CD11b+) was detected after immunization with Flg-HA2-4M2ehs at each point under study. The most pronounced differences were noted 12 h after immunization (*p* = 0.009, *p* = 0.0007).

Changes in the population composition of innate immunity cells in the lungs in response to the introduction of the Flg-HA2-2-4M2ehs protein were accompanied by a significant increase in the expression of the activation marker CD86 on lung cells (CD45+) at all studied points. The results of the analysis of the median fluorescence intensity (MFI) of CD86 are presented in Figure 1B. As early as 12 h after the administration of the recombinant protein, the immunized animals showed an increase in the level of CD86 expression compared with the control group (*p* = 0.03). At later periods, the differences persisted, and there were no pronounced changes in the CD86 level compared to the values obtained 12 h after immunization.

### 3.2. M2e-Specific T-Cell Response in the Lungs of BALB/c and C57Bl6 Mice after Intranasal Immunization

To assess antigen-specific T-cell response after intranasal immunization with the recombinant protein Flg-HA2-2-4M2ehs, lung cells were obtained from five BALB/c mice and six to eight C57Bl6 mice (two trials) two weeks after the third immunization. Lung cells were activated with M2e peptide or the A/H3N2 influenza virus, and the percentage of M2e- and H3N2-specific effector (Tem, CD44+/CD62L-) CD4+ and CD8+ memory T cells, as well as the profile of their cytokines (TNFα and IFN-γ), were estimated.

Upon intranasal administration of the Flg-HA2-2-4M2ehs protein, the formation of M2e-specific effector CD4+ and CD8+ memory T cells was observed in the lungs of BALB/c and C57Bl6 mice (Figure 2). The mean (M ± SEM) of M2e-specific CD4+ Tem and CD8+ Tem in the lungs of BALB /c mice was 0.27 ± 0.12% and 0.14 ± 0.12%, respectively, and was significantly higher than in the control group (0.04 ± 0.05%, 0.02 ± 0.02%, respectively) (Figure 2A). In the lungs of immunized C57Bl/6 mice with the recombinant protein, after activation with the M2e peptide, the proportion of CD8+ (1.3 ± 0.5%) effector memory cells increased significantly (Figure 2A).

The study of the cytokine profile of M2e-specific effector CD4+ and CD8+ memory T cells in the lungs of BALB/c mice with the Flg-HA2-2-4M2ehs protein showed that CD4+ effector memory T cells that produce one type of cytokine CD4+TemTNFα+ (0.26 ± 0.05%) were formed (Figure 3A). At the same time, no significant increase in the number of CD8+ effector memory T cells producing cytokines was observed (Figure 3B). C57Bl6 mice showed only CD8+ effector T cells producing exclusively IFN-γ (1.24 ± 0.55%) (Figure 3B).

### 3.3. A/H3N2-Specific T-Cell Response in the Lungs of BALB/c and C57Bl6 Mice after Intranasal Immunization

Upon activation of lung cells by the A/H3N2 influenza virus, immunized BALB/c mice also showed a significant increase in the population of virus-specific CD4+Tem and CD8+Tem (0.23 ± 0.07%, 0.24 ± 0.09%, respectively) compared with the animals in the control group (0.04 ± 0.02%, 0.02 ± 0.01%, respectively) (Figure 2B). C57Bl6 mice showed a tendency to develop virus-specific CD4+Tem and CD8+Tem (no significant differences were found).

The stimulation of lung cells of immunized BALB/c mice with the influenza A/H3N2 virus caused a significant increase, compared with the control group, in CD4+ and CD8+ Tem producing TNF-α (0.22 ± 0.06 and 0.20 ± 0.1, respectively) (Figure 4A,B). Virus-specific CD8+ Tem monoproducers of IFN-γ (0.93 ± 0.29%) were detected in C57Bl6 mice (Figure 4B).

### 3.4. Expression of the CD107a+ Marker in the Lung Cells of C57Bl/6 Mice after Intranasal Immunization

The activation of mouse lung cells by the M2e peptide or influenza A/H3N2 virus stimulated a significant increase in the expression of the CD107a+ marker on both effector CD8+ and CD4+ memory T cells (Figure 5). The CD107a+ marker was detected in 1.35 ± 0.58% of M2-specific CD4+ Tem and 0.66 ± 0.08% of CD8+ Tem (Figure 5A,B). Among H3N2-specific Tem, CD107a+ expression was detected in 1.33 ± 0.49% of CD4+ Tem and 2.05 ± 0.43% of CD8+ Tem (Figure 5C,D).

Among the antigen-specific CD8+ effector memory cells, the majority of cells (0.67–0.83) simultaneously produced IFN-γ+ and expressed the CD107a+ degranulation marker (Table 1).

## 4. Discussion

For the development of a truly universal influenza vaccine, the neutralizing of anti-HA2 antibodies, non-neutralizing antibodies, T-cell immunity, and anti-NA antibodies are considered equally important correlates of protection [27]. Recent evidence suggests that a balanced induction of multiple correlates of protection against influenza virus antigens is the best way to provide broad cross-protection against different influenza viruses. For many years, the focus in vaccines has been on the production of neutralizing antibodies, but it is becoming increasingly clear that T-cell immunity plays a central role in the fight against respiratory viral infections such as RSV, RV, HMPV, influenza, and coronaviruses [28,29]. Experimental studies in mice and studies of the virus-specific T-cell response in humans have shown that the combination of non-neutralizing antibodies with CD8+ and CD4+ T cells can provide protection against highly pathogenic influenza infection [9,30,31,32,33,34,35]. In a number of viral infections, the presence of T cells with cytotoxic potential correlates with a milder course of the disease [36,37,38]. It should also be noted that the heterosubtypic response and breadth of the protective action can be mediated by T cells directed against conserved influenza antigens [34,39].

The cross-protective properties of the Flg-HA2-2-4M2ehs protein were previously demonstrated [10,11,12] in immunized mice when challenged with viruses of subtypes A/H3N2, A/H7N9, A/H1N1pdm09, A/H2N2, and A/H5N1 at a dose of 5–10 LD50. We have demonstrated that a recombinant protein with two conserved antigens (M2e and HA2–2) induces a strong humoral response and mucosal antibodies (when administered intranasally). Furthermore, the formation of antigen-specific CD4+ T cells in the spleens and lungs, monoproducers of TNF-α or IFN-γ, was observed in BALB/c mice. The mechanism of inducing an immune response to the recombinant protein Flg-HA2-2-4M2ehs through TLR5 was demonstrated using the HEK-Blue™ hTLR5 cell line expressing human TLR5 [11]. In order to study the mechanism of action of the recombinant protein and the formation of immunological memory in more detail, in this work we studied the effect of the recombinant protein on innate immunity cells and on the antigen level of specific effector CD4+ and CD8+ lung T cells.

Our studies showed that the Flg-HA2-2-4M2ehs chimeric protein activated innate immunity factors in the lungs during intranasal immunization, which led to a change in the population composition of innate immunity cells in the lungs of experimental animals in the early stages after immunization (12–48 h). This was manifested in an increase in the relative content of neutrophils, monocytes, and interstitial macrophages, against the backdrop of a decrease in the level of dendritic cells. The decrease in the proportion of DCs (CD11b+ and CD11b−) may be associated with their migration to regional LNs for the presentation of internalized antigens to adaptive immunity cells. Such changes in the cellular composition, as well as increased expression of the CD86 marker, indicate the activating effect of the Flg-HA2-2-4M2ehs protein on the innate immune system, followed by the formation of an antigen-specific adaptive immune response.

Studies of the post-vaccination T-cell immune response during the intranasal immunization of mice with the recombinant protein Flg-HA2-2-4M2ehs were conducted on mice of two lines with different haplotypes: BALB/c (H-2d) and C57Bl/6 (H-2b)—typical representatives of animals with a predominance of Th1 and Th2 types of immune response [40]. In the lungs of BALB/c mice, immunization with the Flg-HA2-2-4M2ehs protein induced the formation of antigen-specific CD4+ and CD8+ effector memory T cells. The study of the cytokine profile demonstrated that they are monoproducers of TNF-α. In mice with a predominance of the Th1 type of immune response (C57Bl/6), the formation of antigen-specific effector CD8+ T cells, predominantly producing IFN-γ+, was shown. These data indicate that the Flg-HA2-2-4M2ehs protein induces the formation of immunological memory of both antigens of the recombinant protein.

We also examined the expression of the CD107a surface antigen on antigen-specific CD4+ and CD8+ effector memory T cells in the lungs after the immunization of mice with the Flg-HA2-2-4M2ehs protein. The data obtained demonstrated the formation of not only CD8+, but also CD4+ effector memory T cells expressing the CD107a antigen after stimulation of lung cells with both antigens, i.e., with cytotoxic potential.

An important role of CD8+ T cells is associated both with the secretion of effector cytokines (IFNγ, TNFα), which are involved in the priming and differentiation of cytotoxic cells, and in the direct destruction of infected and tumor cells [41]. At the same time, the cytotoxic ability of effector CD8+ memory T cells is significantly higher than that of central memory T cells [42]. Cytotoxic T cells (CTL) contain lysosomal granules (perforin, granzyme) in their cytoplasm, ready to degranulate upon activation. These granules have the CD107a degranulation marker, which is transiently expressed on the cell surface when the granules are released. The literature data shows that CD4+ effector T cells, in addition to their auxiliary function (stimulating B-cell maturation and antibody production), also have perforin-mediated cytolytic activity (controls of viral replication) [37,43]. It has been shown that cytotoxic CD4+ T cells are detected after vaccination against smallpox, poliovirus, HIV, and after acute influenza infection [36].

We also found that among the antigen-specific CD8+ effector memory T cells, the majority of cells simultaneously produced IFN-γ+ and expressed the CD107a+ marker, i.e., were polyfunctional. The formation of polyfunctional memory T cells is associated with a milder course of the disease and protection against a lethal influenza infection [44,45,46].

The development of “universal” influenza vaccines based on conserved viral antigens and forming both a humoral and T-cell response aims to minimize the consequences of annual epidemics and provide tools to prevent possible pandemics. T-cell responses induced against highly conserved influenza proteins clear the virus, reduce disease severity, and confer heterosubtypic immunity to influenza viruses [33]. However, there have been concerns that vaccines induced a strong CD8+ response in the lungs may cause damage to lung tissue upon subsequent influenza virus infection [47,48]. A study published recently examined this question [49]. The authors evaluated effects of recombinant adenovirus vectors (rAd) expressing NP of influenza viruses A and B on pulmonary inflammation and function after vaccination and following influenza virus A challenge. It has been shown that the T-cell response in the lung did not cause lung damage upon restimulation and reduced lung damage after influenza infection. These results provide important support for vaccines based on T-cell-mediated protection

## 5. Conclusions

We have previously shown that recombinant protein stimulated the formation of a strong humoral and mucosal response to target antigens and showed a broad protective effect in a challenge experiments. This study demonstrates the effect of the recombinant protein on the factors of innate immunity, which points to the activating effect of the Flg-HA2-2-4M2ehs protein on the innate immune system, followed by the formation of an antigen-specific adaptive immune response. Using the data obtained on mice with different genotypes, it can be concluded that the recombinant protein based on flagellin with conserved antigens of influenza A viruses (M2e and HA2 aa76-130) stimulates the formation of antigen-specific CD4+ and CD8+ mono- and poly-functional effector memory T cells in the lungs. In addition, the importance of the formation of antigen-specific CD4+ and CD8+ effector memory T cells in the lungs with cytotoxic potential (expressing CD107a) should also be noted. Further research will focus on evaluating the effectiveness of the recombinant protein in ferrets as a more suitable animal model for influenza.

## 6. Patent

RU 2757013 C2 RECOMBINANT ANTI-INFLUENZA VACCINE WITH WIDE RANGE OF PROTECTION AND METHOD FOR ITS PREPARATION Application number: RU 2017144741 A (Filing date: 19 December 2017). Publication: 8 October 2021.

## Figures and Tables

**Figure 1 vaccines-10-02098-f001:**
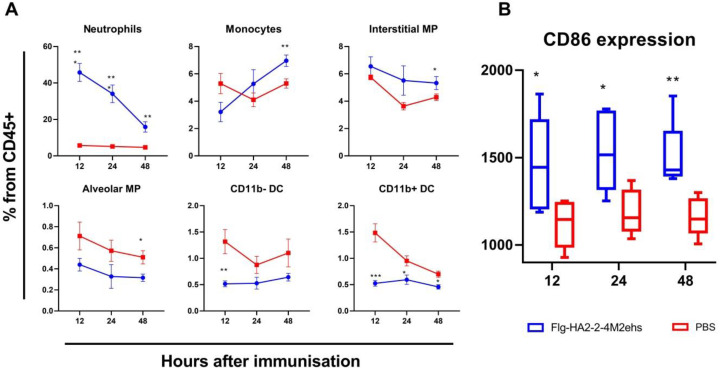
Factors of innate immunity of the lungs of the mice in the early stages (12, 24, 48 h) after intranasal immunization with the chimeric protein Flg-HA2-2-4M2ehs. (**A**) The dynamics of the relative composition of the main populations of innate immunity cells in the lungs of mice at different times after immunization. The graphs show the proportion of different cell populations from the total number of CD45+ cells. (**B**) The dynamics of expression of the CD86 marker in the lungs of mice. The graph shows the values of the median fluorescence intensity (MFI). Significant differences from the control group are marked with * (*: *p* < 0.05; **: *p* < 0.01; ***: *p* < 0.001). The statistical analysis was performed using Student’s *t*-test.

**Figure 2 vaccines-10-02098-f002:**
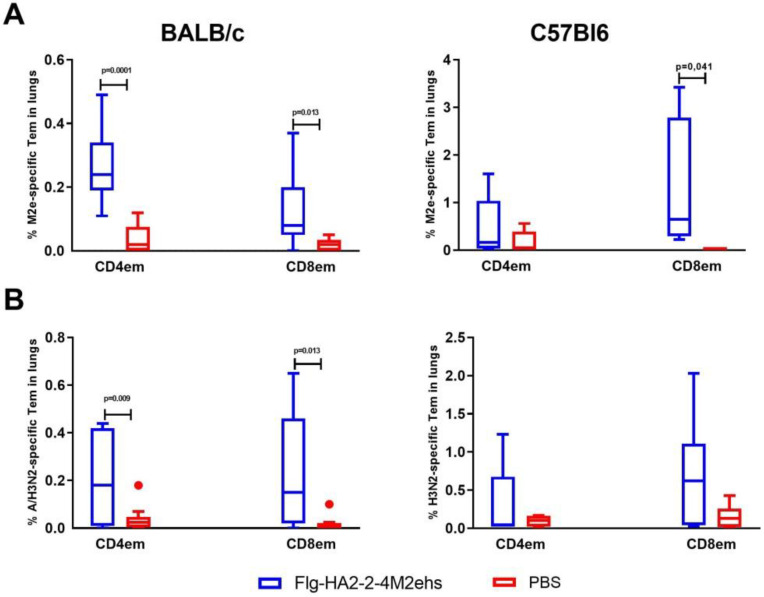
The M2e- (**A**) and A/H3N2-specific (**B**) effector CD4+ and CD8+ memory T cells in lungs after intranasal immunization of BALB/c and C57Bl6 mice with the recombinant protein Flg-HA2-2-4M2ehs. The data are presented as Tukey plots. The data of two experiments (*n* = 8) were summarized. Significant differences from the control group are shown.

**Figure 3 vaccines-10-02098-f003:**
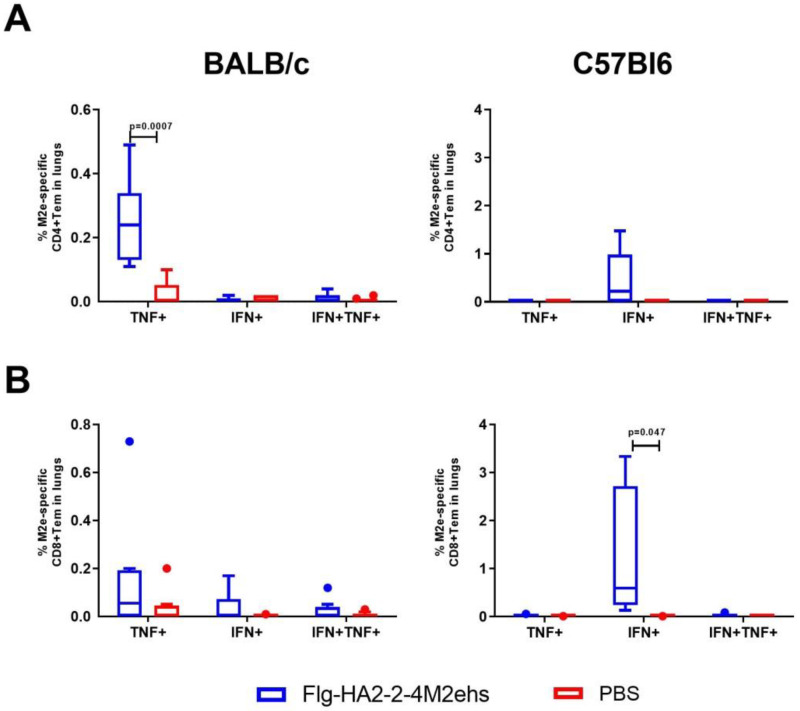
Cytokine profile of M2e-specific CD4+Tem (**A**), CD8+Tem (**B**) in the lungs of BALB/c and C57Bl6 mice after intranasal immunization with recombinant protein Flg-HA2-2-4M2ehs. The data are presented as Tukey plots. The data of two experiments (*n* = 8) were summarized. Significant differences from the control are shown.

**Figure 4 vaccines-10-02098-f004:**
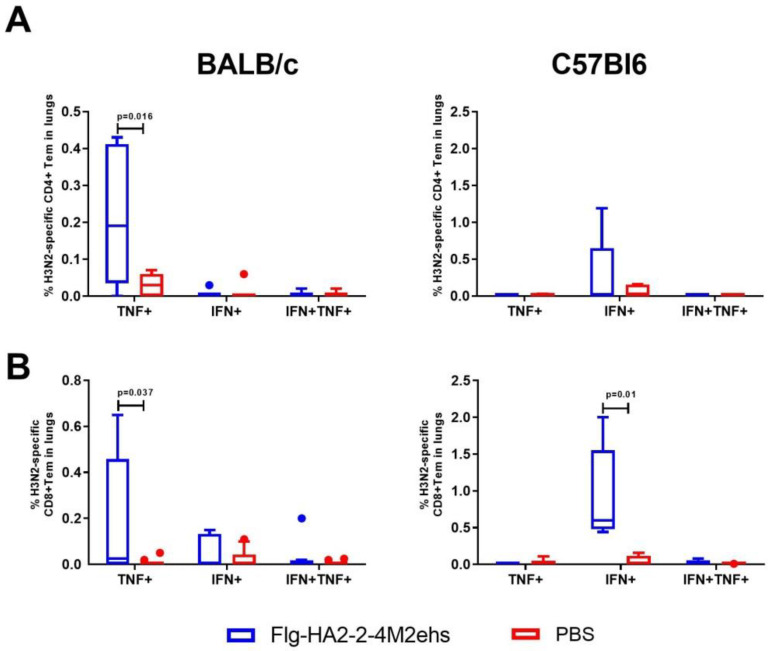
Cytokine profile of H3N2-specific CD4+Tem (**A**), CD8+Tem (**B**) in the lungs of BALB/c and C57Bl6 mice after intranasal immunization with recombinant protein Flg-HA2-2-4M2ehs. The data are presented as Tukey plots. The data of two experiments (*n* = 8) were summarized. Significant differences from the control are shown.

**Figure 5 vaccines-10-02098-f005:**
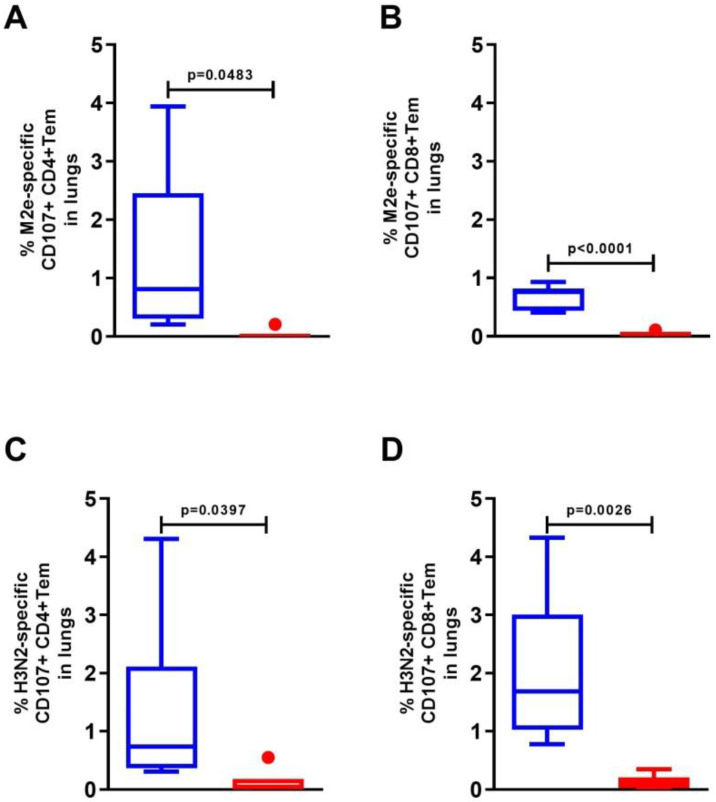
Expression of the degranulation marker CD107a+ on M2e- (**A**,**B**) and A/H3N2-specific (**C**,**D**) effector CD4+ and CD8+ memory T cells in the lungs of C57Bl6 mice after intranasal immunization with recombinant protein Flg-HA2-2-4M2ehs. The data are presented as Tukey plots. The data of two experiments (*n* = 6) were summarized. Significant differences from the control are shown.

**Table 1 vaccines-10-02098-t001:** Functional characteristic of antigen-specific CD8+Tem lungs cells in C57Bl/6 mice.

	Mouse Number					
	1	2	3	4	5	6
*M2e-specific CD8+Tem*						
CD107a+	+	+	+	+	−	+
IFN-γ+	−	++	++	−	+	−
IFN+CD107a+	+	+	−	+	+	+
*H3N2-specific CD8+Tem*						
CD107a+	++	++	+	++	+	++
IFN-γ+	−	++	++	−	−	+
IFN+CD107a+	+	+	−	−	+	+

++ more than 1% of cells, + 0.1–1.0% of cells, − less than 0.1% of cells.

## Data Availability

Data are available through contact with the correspondent author.

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
