# Peer review of "Flagellin-Fused Protein Targeting M2e and HA2 Induces Innate and T-Cell Responses in Mice of Different Genetic Lines"

_vaccines, 2022, doi:10.3390/vaccines10122098_

Round 1

Reviewer 1 Report

The authors intranasally immunized two different mice strains with a flagellin based recombinant protein (Flg-HA2–2-4M2ehs) to evaluate its antigenicity. The authors used flow cytometry to analyze both the innate and the adaptive immunity response after immunization. As a result, the recombinant protein can induce both innate and antigen-specific T cell response in the lungs of experiment animals. I think the manuscript is practical and should be accepted. The results presented in the manuscript can be useful for the vaccine development field. It would be ideal if the authors could test whether the immunization can also generate B cell response and protect mice against influenza infection. But this is beyond the scope of this paper.

Some minor issues:

1.     The authors should also talk about different cell markers they used in this study in the introduction section, e.g. CD86, CD107a+. For people not in the immunology field it could be hard to follow.

2.     I recommend putting the abbreviations section before introduction so it’s easier to read.

Author Response

Manuscript ID: vaccines-2007828:

Flagellin-fused protein targeting M2e and HA2 induces innate and T-cell responses in mice of different genetic lines

Reviewer 1

Comments and Suggestions for Authors

The authors intranasally immunized two different mice strains with a flagellin based recombinant protein (Flg-HA2–2-4M2ehs) to evaluate its antigenicity. The authors used flow cytometry to analyze both the innate and the adaptive immunity response after immunization. As a result, the recombinant protein can induce both innate and antigen-specific T cell response in the lungs of experiment animals. I think the manuscript is practical and should be accepted. The results presented in the manuscript can be useful for the vaccine development field. It would be ideal if the authors could test whether the immunization can also generate B cell response and protect mice against influenza infection. But this is beyond the scope of this paper.

Response to comment

Thank you very much for your comment and questions. In our earlier work [Tsybalova, L.M.; Stepanova, L.A.; Shuklina, M.A.; Mardanova, E.S.;  Kotlyarov, R.Y.; Potapchuk, M.V.; Petrov, S.A.; Blokhina, E.A.; Ravin, N.V. Combination of M2e peptide with stalk HA epitopes of influenza A virus enhances protective properties of recombinant vaccine. PLoS One 2018, 13, e0201429. Stepanova, L.A.; Mardanova, E.S.; Shuklina, M.A.; Blokhina, E.A.; Kotlyarov, R.Y.;  Potapchuk, M.V.; Kovaleva, A.A.; Vidyaeva, I.G.; Korotkov, A.V.; Eletskaya, E.I.; Ravin, N.V.; Tsybalova, L.M.  Flagellin-fused protein targeting  M2e and HA2  induces  potent  humoral  and T-cell respones  and  protects  mice  against various influenza viruses a subtypes. J. Biomed. Sci. 2018, 25, e33], we demonstrated that both intranasal and subcutaneous immunization with Flg-HA2–2-4M2ehs of mice BALB/c stimulated the formation of a strong antigen-specific humoral systemic and local immune response. We also showed that immunized mice BALB/c were almost completely protected from infection with influenza viruses of subtype A/H1N1, A/H3N2, A/H7N9, H5N1.

Some minor issues:

  1. The authors should also talk about different cell markers they used in this study in the introduction section, e.g. CD86, CD107a+. For people not in the immunology field it could be hard to follow.

Response: Corrected

The section 2.5 was supplemented with a description of markers of innate immunity cells “This panel enabled the alveolar macrophages (MHCII+CD64+ CD11c+CD11b-), interstitial macrophages (MHCII+CD64+CD11b+CD11c+/-), monocytes (MHCII-CD64+CD24+), neutrophils (SSChiCD45+Ly6G+) and two dendritic cell populations: DC1 CD11b- (CD45+CD11c+CD11b- MHCII+CD64-CD24+) and DC2 CD11b + (CD45+CD11c+CD11b+MHCII+CD64-CD24+) to be identified.”  

The section 2.6 was supplemented with a description of markers of T-memory cells “The formation of an antigen-specific T-cell response in the lungs of mice was assessed by the number of effector memory CD4+ and CD8+ T-cells (CD44+/CD62L-) and their functional characteristics by the expression of a degranulation marker (CD107+) and the production of intracellular cytokines (TNFα and IFN-γ).”

  1. I recommend putting the abbreviations section before introduction so it’s easier to read.

Response:

The abbreviations is located after the main text, as usual in the journal

Reviewer 2 Report

Mutations in viral genome constantly produce new variants or strains that can potentially evade the existing effective immune response, thereby creating new threats to the global public health. Influenza, among many viruses, remains a public health threat despite the ongoing developments of flue vaccines. Although vaccinations that trigger B cell responses and elicit neutralizing antibodies provide the most effective protection against viral infections, influenza viruses carrying mutated HA1 could lead to infective vaccines against an upcoming flu season. On the other hand, vaccines or natural infections that trigger responses against the most conserved regions on the viral surface could provide broader and longer-lasting protection against a broad panel of viral strains. The authors focused on this direction to use the comparatively conserved HA2, which is usually shielded by HA1, and M2 ecto-domain, which is conserved yet less accessible due to the dense packing of HA spikes on the viral surface, as their vaccine candidate, to activate the innate immune responses followed by certain level of adaptive immune response. The data presented in this work clearly showed the effective activation of CD4+/CD8+ T cell responses, which could serve as informative evidence for researchers working on T cell vaccine designs. 

The data are solid and clear. My biggest concern remains at the real-world effectiveness of this type of vaccines. It is certainly helpful to stimulate the body in recognizing the more conserved viral epitopes such that an initial immune response should be effectively triggered against a previously unrecognized novel variant or strain, how much protection can be provided in preventing severe pathological effects remains a debate. The authors could probably provide some of their opinions in the Discussion. In addition, the authors could discuss the protective effects of the stronger T cell response triggered by the recent mRNA vaccines, thereby help prove the necessity for the development of vaccines such as this one. 

Author Response

Manuscript ID: vaccines-2007828:

Flagellin-fused protein targeting M2e and HA2 induces innate and T-cell responses in mice of different genetic lines

Reviewer 2

Comments and Suggestions for Authors

Mutations in viral genome constantly produce new variants or strains that can potentially evade the existing effective immune response, thereby creating new threats to the global public health. Influenza, among many viruses, remains a public health threat despite the ongoing developments of flue vaccines. Although vaccinations that trigger B cell responses and elicit neutralizing antibodies provide the most effective protection against viral infections, influenza viruses carrying mutated HA1 could lead to infective vaccines against an upcoming flu season. On the other hand, vaccines or natural infections that trigger responses against the most conserved regions on the viral surface could provide broader and longer-lasting protection against a broad panel of viral strains. The authors focused on this direction to use the comparatively conserved HA2, which is usually shielded by HA1, and M2 ecto-domain, which is conserved yet less accessible due to the dense packing of HA spikes on the viral surface, as their vaccine candidate, to activate the innate immune responses followed by certain level of adaptive immune response. The data presented in this work clearly showed the effective activation of CD4+/CD8+ T cell responses, which could serve as informative evidence for researchers working on T cell vaccine designs.

The data are solid and clear. My biggest concern remains at the real-world effectiveness of this type of vaccines. It is certainly helpful to stimulate the body in recognizing the more conserved viral epitopes such that an initial immune response should be effectively triggered against a previously unrecognized novel variant or strain, how much protection can be provided in preventing severe pathological effects remains a debate. The authors could probably provide some of their opinions in the Discussion. In addition, the authors could discuss the protective effects of the stronger T cell response triggered by the recent mRNA vaccines, thereby help prove the necessity for the development of vaccines such as this one.

Response to comment

Thank you very much for your comment.

In the discussion section, we added: “The development of "universal" influenza vaccines based on conserved viral antigens and forming both a humoral and T-cell response aims to minimize the consequences of annual epidemics and have tools to prevent possible pandemics. T-cell responses induced against highly conserved influenza proteins clear the virus, reduce disease severity, and confer heterosubtypic immunity to influenza viruses [33]. However, there have been concerns that vaccines induced a strong CD8+ response in the lungs may cause damage to lung tissue upon subsequent influenza virus infection [48, 49]. A study published recently examined this question [50].  The authors evaluated effects of recombinant adenovirus vectors (rAd) expressing NP of influenza viruses A and B on pulmonary inflammation and function after vaccination and following influenza virus A challenge. It has been shown that the T-cell response in the lung did not cause lung damage upon restimulation and reduced lung damage after influenza infection. These results provide important support for vaccines based on T cell-mediated protection.”

Reviewer 3 Report

Comments on vaccines-2007828

Flagellin-fused protein targeting M2e and HA2 induces innate and T-cell responses in mice of different genetic lines 

In the present study, Liudmila A. Stepanova et al. investigated the immune responses induced by a recombinant M2e and HA2 protein fused with flagellin to be used as a broad-spectrum vaccine candidate in mouse models. They showed that the vaccine candidate was able to induce antigen-specific CD4+ and CD8+ effector memory T cells (expressing CD107a) with cytotoxic potential. The study is interesting and potentially useful, but several major concerns should be addressed. 

Major concerns:

1.              Specific antibodies induced by the vaccine candidate should be examined.

2.              Cross-neutralization and hemagglutination-inhibition tests and virulent challenge experiments should be carried out to demonstrate the broad-spectrum protection potential of the vaccine candidate.

3.              The presentation of the figures and tables can be improved.

4.              The manuscript writing should be modified by native English speakers.

Author Response

Manuscript ID: vaccines-2007828:

Flagellin-fused protein targeting M2e and HA2 induces innate and T-cell responses in mice of different genetic lines

Reviewer 3

Flagellin-fused protein targeting M2e and HA2 induces innate and T-cell responses in mice of different genetic lines

In the present study, Liudmila A. Stepanova et al. investigated the immune responses induced by a recombinant M2e and HA2 protein fused with flagellin to be used as a broad-spectrum vaccine candidate in mouse models. They showed that the vaccine candidate was able to induce antigen-specific CD4+ and CD8+ effector memory T cells (expressing CD107a) with cytotoxic potential. The study is interesting and potentially useful, but several major concerns should be addressed.

Major concerns:

  1. Specific antibodies induced by the vaccine candidate should be examined.

Response:

In our earlier work [Tsybalova, L.M.; Stepanova, L.A.; Shuklina, M.A.; Mardanova, E.S.;  Kotlyarov, R.Y.; Potapchuk, M.V.; Petrov, S.A.; Blokhina, E.A.; Ravin, N.V. Combination of M2e peptide with stalk HA epitopes of influenza A virus enhances protective properties of recombinant vaccine. PLoS One 2018, 13, e0201429. Stepanova, L.A.; Mardanova, E.S.; Shuklina, M.A.; Blokhina, E.A.; Kotlyarov, R.Y.;  Potapchuk, M.V.; Kovaleva, A.A.; Vidyaeva, I.G.; Korotkov, A.V.; Eletskaya, E.I.; Ravin, N.V.; Tsybalova, L.M.  Flagellin-fused protein targeting  M2e and HA2  induces  potent  humoral  and T-cell respones  and  protects  mice  against various influenza viruses a subtypes. J. Biomed. Sci. 2018, 25, e33], we demonstrated that both intranasal and subcutaneous immunization with Flg-HA2–2-4M2ehs of mice BALB/c stimulated the formation of a strong antigen-specific humoral systemic and local immune response.

  1. Cross-neutralization and hemagglutination-inhibition tests and virulent challenge experiments should be carried out to demonstrate the broad-spectrum protection potential of the vaccine candidate.

Response:

It is not possible to set up a microneutralization reaction and hemagglutination-inhibition tests, since the first subunit of HA is not included in the recombinant protein, and antibodies formed to conserved antigens are not neutralizing. The broad-spectrum protection potential of the vaccine candidate has been demonstrated in our earlier work [Tsybalova, L.M.; Stepanova, L.A.; Shuklina, M.A.; Mardanova, E.S.;  Kotlyarov, R.Y.; Potapchuk, M.V.; Petrov, S.A.; Blokhina, E.A.; Ravin, N.V. Combination of M2e peptide with stalk HA epitopes of influenza A virus enhances protective properties of recombinant vaccine. PLoS One 2018, 13, e0201429. Stepanova, L.A.; Mardanova, E.S.; Shuklina, M.A.; Blokhina, E.A.; Kotlyarov, R.Y.;  Potapchuk, M.V.; Kovaleva, A.A.; Vidyaeva, I.G.; Korotkov, A.V.; Eletskaya, E.I.; Ravin, N.V.; Tsybalova, L.M.  Flagellin-fused protein targeting  M2e and HA2  induces  potent  humoral  and T-cell respones  and  protects  mice  against various influenza viruses a subtypes. J. Biomed. Sci. 2018, 25, e33].  We showed that immunized mice BALB/c were almost completely protected from infection with influenza viruses of subtype A/H1N1, A/H3N2, A/H7N9, H5N1.

  1. The presentation of the figures and tables can be improved.

Response: Thank you for your comment, but we do not see a way to improve the the figures and tables.

  1. The manuscript writing should be modified by native English speakers.

Response: The manuscript was edited by a native English speaker.

Round 2

Reviewer 3 Report

The manuscript needs extensive editing and formatting (e.g. subscript and superscript, etc.).

Author Response

Manuscript ID: vaccines-2007828:

Flagellin-fused protein targeting M2e and HA2 induces innate and T-cell responses in mice of different genetic lines

 Reviewer 3

Comments and Suggestions for Authors

The manuscript needs extensive editing and formatting (e.g. subscript and superscript, etc.).

Response:

Thank you very much for reading the manuscript and for your comments. The manuscript has been edited and formatted.